# Risk Management for Mitigating Benchmark Failure Modes: BenchRisk

**Sean McGregor,**[1,2,*] **Victor Lu,**[3,†] **Vassil Tashev,**[3,†] **Armstrong Foundjem,**[4,‡]
**Aishwarya Ramasethu,**[5,‡] **Mahdi Kazemi,**[10,‡] **Chris Knotz,**[3,‡] **Kongtao Chen,**[6,‡]
**Alicia Parrish,**[7,∘] **Anka Reuel,**[8,¶] **Heather Frase**[9,2,¶]

[1]AI Verification and Evaluation Research Institute,
[2]Responsible AI Collaborative, [3]Independent, [4]Polytechnique Montreal, [5]Prediction Guard,
[6]Google, [7]Google Deepmind, [8]Stanford University, [9]Veraitech, [10]University of Houston

Contribution equivalence classes (∗, †, ‡, ∘, ¶) detailed in acknowledgments

## Abstract

Large language model (LLM) benchmarks inform LLM use decisions (e.g., "is this LLM safe to deploy for my use case and context?"). However, benchmarks may be rendered unreliable by various failure modes that impact benchmark bias, variance, coverage, or people's capacity to understand benchmark evidence. Using the National Institute of Standards and Technology's risk management process as a foundation, this research iteratively analyzed 26 popular benchmarks, identifying 57 potential failure modes and 196 corresponding mitigation strategies. The mitigations reduce failure likelihood and/or severity, providing a frame for evaluating "benchmark risk," which is scored to provide a metaevaluation benchmark: BenchRisk. Higher scores indicate that benchmark users are less likely to reach an incorrect or unsupported conclusion about an LLM. All 26 scored benchmarks present significant risk within one or more of the five scored dimensions (comprehensiveness, intelligibility, consistency, correctness, and longevity), which points to important open research directions for the field of LLM benchmarking. The BenchRisk workflow allows for comparison between benchmarks; as an open-source tool, it also facilitates the identification and sharing of risks and their mitigations.

## 1  Introduction

Benchmarks have played a central role in the rapid advancement of large language models (LLMs), both in terms of driving their capabilities and capturing their risks (Srivastava et al. [2023]). Now with a wealth of new use cases supported by general-purpose models, LLM benchmark authors are proposing to evidence safety and regulatory decisions (e.g., ML Commons [2024], Guldimann et al. [2025], Zeng et al. [2024]). However, users are hesitant to rely on current benchmarks for real-world decisions (Hardy et al. [2025]), including those presented by frontier model release documentation (Röttger et al. [2024], Bommasani et al. [2024]). Skepticism of benchmark use outside the research and development communities is well-founded. Previous research has identified broad types of benchmark deficiencies, as outlined in Section 2. This work treats benchmark deficiencies as a tool for benchmarking benchmark reliability. Through an iterative process analyzing 26 benchmarks, we collected and classified 57 LLM benchmark failure modes (Definition 1) with a corresponding set of 196 mitigations. Proceeding within the context of a risk management framework, we produced a benchmark reliability benchmark to *1)* help benchmark users know when they should avoid relying on a benchmark, *2)* assist benchmark authors in prioritizing failure mode mitigations, *3)* motivate

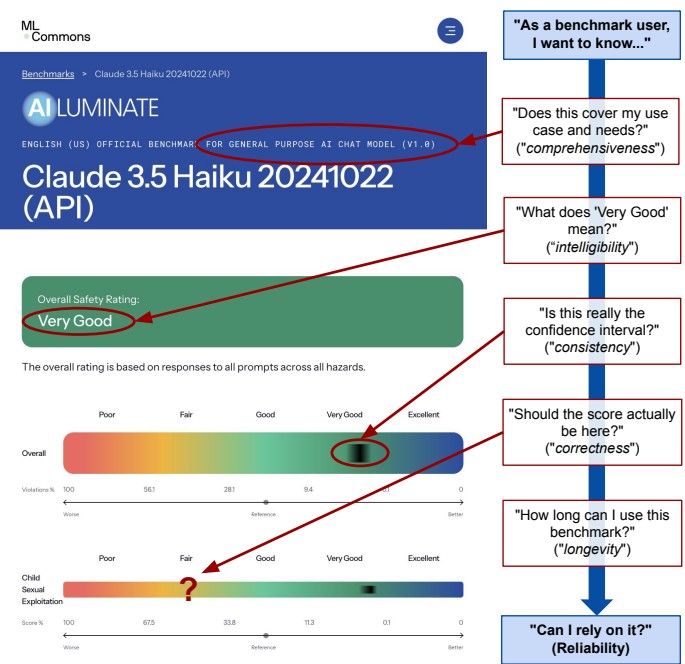

Figure 1: The five dimensions of benchmark reliability (comprehensiveness, intelligibility, consistency, correctness, and longevity) mapped to a user's decision-making process. The questions illustrate how a user determines overall reliability ("Can I rely on it?") from an interface like AILuminate (ML Commons [2024]).

additional research in making benchmarks more reliable, *4)* support the comparison of different benchmarks according to their reliability.

> **Definition 1: Benchmark Failure Mode**
>
> The way in which a benchmark could potentially provide the user with faulty real-world decision-making information. *Adapted from National Security Agency [2015] and Rausand and Høyland [2004].*

Users who rely on benchmarks that exhibit failure modes (e.g., by making a decision about what is a "safe" use case for a specific model) may arrive at unsupported or erroneous deployment decisions, potentially leading to real-world harm. Although benchmarks may serve an important informational purpose for understanding and comparing LLMs, benchmark users lack a means of understanding the reliability of benchmarks without dedicating considerable time and resources to evaluating each benchmark. The aim of this paper is to close this gap.

The framework for benchmark reliability is based on the evaluation of failure risk. For users, it is difficult to understand how failure modes may render a benchmark unreliable (see Figure 1). For benchmark developers, it is similarly difficult to identify and prioritize risks posed by different benchmark design, development, and operational decisions and select mitigation strategies that balance risks and benefits. Just as the only way to ensure an airplane doesn't crash is to never leave the ground, the only benchmark that is always 100 percent reliable is one that is never used. Assessing benchmark reliability requires a means to reason about priorities and allocating resources accordingly. Risk management processes enable structured reasoning for such a triage process. To address the multiplicity of benchmark failures, we take inspiration from the reliability engineering community, which explicitly models failures of both the technology (e.g., a plane) and the human factors (e.g., its pilot) to estimate risk.

We are concerned with measuring and reporting on the questions of Figure 2 for benchmark users and for providing a means of efficiently aligning benchmark development to minimize risks implied by those questions. A benchmark that a user does not understand is not reliable for that user,

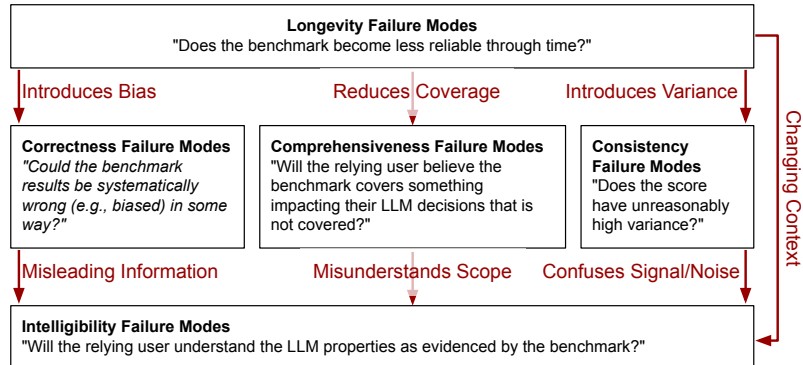

Figure 2: Assessed reliability dimensions and their relationships. Longevity failures degrade a benchmark over time, impacting correctness, comprehensiveness, and consistency. These properties, in turn, are foundational to a user's ability to understand the results (intelligibility).

which places benchmark and documentation "intelligibility" downstream from bias, variance, and coverage properties. These properties degrade through time (e.g., when developers train to the evaluation set). The user-centered dimensions and their relationship to concepts found within the research communities of reliability engineering, information security, and statistical validity are further explored in Section 2.

We note that many benchmarks are not produced for the purpose of evidencing real-world decisions. These benchmarks can still be groundbreaking for scientific and optimization purposes, but we show how they could be changed to support real-world decision-making. Regardless of benchmark author intent, benchmarks are often reported for commercial products (Röttger et al. [2024]).

In this paper, we're estimating benchmark reliability risks by combining the efforts of benchmark validity research (e.g., Wallach et al. [2025]) with interventions from reliability engineering and information security risk management (e.g., National Institute of Standards and Technology [2012]). Processing each benchmark's publicly available documentation in turn, we captured failure modes risking *benchmark reliability*.

> **Definition 2: Benchmark Reliability**
>
> The ability for a benchmark to inform real-world decision-making in a stated operating context for a specified amount of time and with no failures.

We scored the failure modes based on expert estimates of their severity and identified potential mitigations. Similarly, we used expert judgment to estimate each mitigation's reduction in risk severity or likelihood. Going forward, these estimates will be subject to a dynamic community-driven adjustment process to update and improve the scores over time. A higher score in our framework indicates that the scored benchmark mitigates more risk and thus may be more reliable for evidencing real-world decision-making. We followed an iterative process (see Appendix B) of reading benchmark documentation for 26 benchmarks to produce a list of 57 failure modes and 196 mitigations.

The contributions of this work are as follows:

*1)* We present a risk management process for designing more reliable benchmarks.

*2)* We score 26 benchmarks to identify the state of practice of reliable benchmarking: **BenchRisk**

*3)* We provide a means for benchmark authors to update scores and submit additional benchmarks by revealing additional details not apparent to outside assessment

*4)* We publish all the materials associated with this process (see `https://benchrisk.ai/`)

*5)* We provide infrastructure, including a publicly accessible GitHub repository (BenchRisk Team [2024]), to establish a community-driven consensus process to identify, share, mitigate, and score benchmarks according to their reliability properties.

Mitigations increasing benchmark reliability may reduce the benchmark's scientific utility. For example, a benchmark that is downloadable by researchers may advance the state of the art in LLM capabilities more quickly than a benchmark that is more private for the purpose of protecting its longevity. Therefore, we do not believe that all benchmarks should be reformed per our mitigations, but those that seek to inform real-world decisions should seek to mitigate the failure modes we identify.

## 2 Related work

AI safety benchmarks are being used to influence decision making (Röttger et al. [2024], Bommasani et al. [2024]), but the reliability of these benchmarks is being called into question, for example as a result of developers training (intentionally or inadvertently) on test data (Li et al. [2024], Magar and Schwartz [2022], Zhou et al. [2023], Balloccu et al. [2024]), unclear benchmark scope (Raji et al. [2021]), application of benchmarks contrary to their published purposes, etc. (see, e.g., Liao and Xiao [2023], McIntosh et al. [2024], Banerjee et al. [2024], Roose [2024], Hardy et al. [2025], Keegan [2024], Anthropic [2024]).

Towards improving the scientific reliability of LLM benchmarks, a variety of recent studies examined the gaps in benchmark quality giving rise to these issues. BetterBench from Reuel et al. [2024] focused on how well a benchmark was designed, implemented, documented and how well it will be maintained. BetterBench adopts the definition of Raji et al. [2021] for "benchmark," which calls benchmarks "...a particular combination of a dataset or sets of datasets (at least test data, sometimes also training data), and a metric, conceptualized as representing one or more specific tasks or sets of abilities, **picked up by a community of researchers as a shared framework for the comparison of methods.**" BenchRisk changes the bolded text to "consumed by users to inform their decision making." The modification substantially expands the responsibilities of benchmark authors to examine how reliably a benchmark informs a decision maker of model properties of interest.

The definitions differ on who's using the benchmark (researcher vs. end-user) as much as it does on the purpose (measuring scientific progress vs. supporting decision making). For a researcher to progress the science of training LLMs, their chosen benchmarks must be well documented and widely shared so advances in the state of the art will be comparable. Instance-level data availability inform where the model might be improved through either model architecture or training set changes. However, while sharing supports transparency and replicability it also increases the contamination risk of future systems under test ("SUTs", i.e., the system being benchmarked), resulting in a gap between benchmark scores and actual capabilities (Haimes et al. [2024]). Still, the objectives of measuring research progress and supporting real-world decision making are only partially in tension. The audience and aim of researchers diverge in the BenchRisk dimension scoring temporal failure modes related to the longevity of the benchmark, but they are aligned in the other four of Figure 2.

Benchmark reliability for each of the dimensions (comprehensiveness, intelligibility, consistency, correctness, and longevity, see Figures 1 and 2) can be enhanced by the adoption of best practices introduced in other works (e.g., Reuel et al. [2024], Cao et al. [2025]). These works identify what should be done, but the risks these best practices are addressing remain informal without quantifying risk in terms of likelihood (Definition 3) and severity (Definition 4). Such a framing is required for understanding real-world, use-case specific risks to benchmark reliability and for triaging and prioritizing corresponding risk mitigation efforts.

> **Definition 3: Likelihood.**
>
> A factor based on a subjective estimate of the probability that a given failure mode will materialize and impact reliability. *Adapted from National Security Agency [2015] and Rausand and Høyland [2004]*)

> **Definition 4: Severity**
>
> An assessment of the relative consequence of mitigating/remediating the failure mode. *Adapted from National Institute of Standards and Technology [2012]*

Table 1: The severity rankings with descriptions. These are adapted from United States of America Department of Defense [2012], which establish four levels of severity.

| Severity | Interpretation |
|---|---|
| **<= 1.00** | *catastrophic*: Could result in the immediate irreversible full loss of utility of the benchmark |
| **< 0.75** | *critical*: Could result in significant reduction in a benchmark's comprehensiveness, intelligibility, consistency, correctness, or longevity. |
| **< 0.50** | *degraded*: Could result in moderate reduction in a benchmark's comprehensiveness, intelligibility, consistency, correctness, or longevity. |
| **< 0.25** | *marginally degraded*: Could result in minor reduction in at least one benchmark dimension of comprehensiveness, intelligibility, consistency, correctness, or longevity. |

The reliability analysis frame extends beyond statistical consistency to encompass broader systemic considerations (see: Rausand and Høyland [2004], McLinn [2011]). In safety-critical domains like aviation, reliability is not defined by singular outcomes but by the capacity of a system to prevent harm under uncertain and evolving conditions. For example, while pilot error is often cited as a cause of accidents, safety engineering instead seeks to trace such failures to latent factors – design flaws, inadequate training, or insufficient warning systems – underscoring the importance of systems that anticipate and mitigate foreseeable risks. In the context of AI, and particularly in the use of LLM benchmarks, similar principles apply. Misleading or incomplete benchmark results can lead to inappropriate deployment decisions, with serious downstream consequences. However, benchmark quality is often treated narrowly, focused on reproducibility or statistical rigor alone. This underappreciates the complexity of how benchmark evidence is generated, interpreted, and applied. BenchRisk expands the scope of reliability to include dimensions such as comprehensiveness, intelligibility, and longevity – reflecting the need for benchmarks to actively mitigate risks of misinterpretation or misuse.

## 3  Risk Assessment with BenchRisk

Risk assessment is a process that determines possible failure modes, along with their likelihood and consequences (Rausand and Haugen [2020]). Such assessments help decision makers develop mitigations and formulate response priorities. They are used as a tool across sectors (e.g., International Organization for Standardization [2018]) to elicit in-house severity and likelihood values for identified risks, which can then be used to score the total risk faced by an organization. Such analyses are established in the context of NIST information security practices, but they have yet to be adopted for AI evaluation risks. Our work is aiming to bridge this gap by applying an external, structured means of risk scoring benchmarks as outside parties, from which the benchmark authors may subsequently engage in a consensus process to refine those scores (see Appendix A).

In adapting the NIST framework, BenchRisk replaces the "threats" of information security with the "failure modes" of reliability engineering. "Threats" implies the involvement of a threat actor (e.g., a company working to exploit a benchmark). "Failure mode" aligns with reliability engineering and doesn't presume the existence of a threat actor. Each failure mode represents a condition under which a user might misinterpret benchmark results, potentially leading to unsupported or harmful conclusions about an LLM's fitness for a given application context. Benchmarks receive higher BenchRisk scores when they demonstrate strong, targeted mitigations to such failure modes.

Established information security practices (National Institute of Standards and Technology [2012]) require explicitly specifying the purpose, scope, assumptions, information sources, and analysis models. A corresponding specification for BenchRisk can be found in Appendix B.

For the purpose of BenchRisk, we differentiate severities according to the levels of Table 1. This approach for severity rankings is a common risk assessment process across sectors and is found in systems reliability theory (Rausand and Høyland [2004]), information security (National Institute of Standards and Technology [2012]), and for natural disasters (Caldera and Wirasinghe [2021])

Risk assessors do not define a likelihood function in the statistical sense. Rather, they assign a likelihood score (or risk level) based on available evidence, expert judgment, and professional experience.

For BenchRisk, all failure modes are assigned an initial likelihood of 1.0. The assumed starting likelihood takes a worst case viewpoint; the likelihood may be reduced for each benchmark, depending on mitigations implemented by benchmark authors (see Algorithm 1). Benchmark authors can then score points by mitigating severity or likelihood, which jointly determine "risk."

---

**Definition 5: Risk to Benchmark Reliability**

A composite measure of a failure mode's probability of occurring and the magnitude or degree of the consequences of the corresponding failure. *Adapted from National Institute of Standards and Technology [2024]). BenchRisk expresses risk as $(severity * likelihood)$, as is commonplace in risk management.*

---

**Definition 6: Risk Mitigation**

Accepting, avoiding, reducing, sharing, or transferring risk. *From Raji et al. [2021].*

---

Mitigations reduce either the failure mode severity, the failure mode likelihood, or both. The risk reduction is aggregated for each reliability dimension for each benchmark, which is shown as the BenchRisk score. Stated formally, let $d$ be a reliability dimension within the set of reliability dimensions defined in Figure 2. A dimension $d$ is degraded by failure mode $f$ in the set of failure modes $F_d$. Each failure mode has a severity $f_s \in [0, 1]$ and an associated likelihood $f_l$ of 1.0 prior to any mitigation(s) implemented. Mitigation $m$ is among the set of possible mitigations $M_{d,f}$ to failure mode $f$ and it reduces a failure mode's likelihood by $m_l$ and severity by $m_s$. Each mitigation stacks, such that if each of two mitigations reduces a failure mode's likelihood by 0.5, the resulting likelihood is 0.25. The calculation for BenchRisk is now given in Algorithm 1 and several example calculations are given in Figure 3.

---

**Algorithm 1** BenchRisk for dimension $d$

1: Initialize $F_d \leftarrow [\{$ failure modes to dimension d $\}]$
2: Initialize $M_d \leftarrow [\{$ adopted mitigations to $F_d$ $\}]$
3: Initialize $score \leftarrow 0.0$
4: **for all** $f \in F_d$ **do**
5:     $likelihood \leftarrow 1.0$
6:     $severity \leftarrow f_s$
7:     **for all** $m \in M_{d,f}$ **do**
8:         $likelihood \leftarrow likelihood - likelihood \times m_l$
9:         $severity \leftarrow severity - severity \times m_s$
10:     **end for**
11:     $score \leftarrow score + |(likelihood \times severity) - (f_l \times f_s)|$
12: **end for**
13: **return** $score$

---

We seeded BenchRisk iteratively, from the ground up, by processing a series of benchmark research papers and their supporting documentation. We selected benchmarks to score from the BetterBench list of models along with several arbitrarily chosen by co-authors based on professional interest. At each iteration, new failure modes and mitigations were identified, added, and scored across the growing collection of benchmarks. All scores presented within this work were subject to a primary and secondary reviewer, who discussed and eventually reached agreement on the appropriateness of affirming a mitigation given publicly known information about each benchmark. Reaching agreement sometimes involved clarifying descriptions of failure modes and their mitigations, which were captured and applied for all scores. The complete set of risks and mitigations are available via appendices A and B along with additional resources detailing the initial set of failure modes and mitigations.

**Failure Mode #46 (Longevity):**
*Developers can run the benchmark an unlimited number of times*
(Severity 0.8)*(Likelihood 1.0) = 0.8 Points

**Failure Mode #25 (Correctness):**
*Developers place evaluator or other test ground truth within system chain*
(Severity 0.9)*(Likelihood 1.0) = 0.9 Points

**Mitigation #67:** *Do you restrict or avoid evaluation on demand to preserve benchmark integrity?*

| No | Yes |
|----|-----|

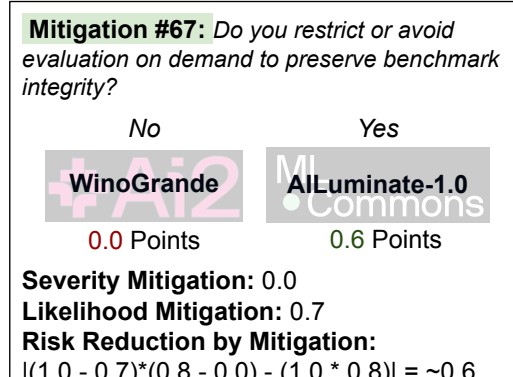

| **WinoGrande** | **AILuminate-1.0** |
|----------------|--------------------|
| 0.0 Points | 0.6 Points |

**Severity Mitigation:** 0.0
**Likelihood Mitigation:** 0.7
**Risk Reduction by Mitigation:**
|(1.0 - 0.7)*(0.8 - 0.0) - (1.0 * 0.8)| = ~0.6

**Mitigation #28:** *Do you refrain from making the evaluator or ground truth publicly available?*

| **WinoGrande** | **AILuminate-1.0** |
|----------------|--------------------|
| 0.0 Points | 0.7 Points |

**Mitigation #93:** *Is the evaluator strictly algorithmic (i.e., applying a list of correct answers) with no legitimate reason to be embedded in the system-under-test (SUT) chain?*

| **WinoGrande** | **AILuminate-1.0** |
|----------------|--------------------|
| 0.8 Points | 0.0 Points |

Figure 3: Example risk reduction calculations for two benchmarks against two different failure modes. **Left (Longevity)**: AILuminate-1.0 gains 0.6 points by applying Mitigation 67, which reduces the likelihood of Failure Mode 46. The "severity" of failure mode 46 is not reduced by mitigation 67 because severity and likelihood are considered separately. **Right (Correctness)**: WinoGrande and AILuminate-1.0 apply different mitigations for Failure Mode 25, earning 0.8 and 0.7 points respectively. Additional failure modes and mitigations are available at *BenchRisk.ai*

After applying BenchRisk, five co-authors not including the most frequent secondary reviewer separately scored the BBQ benchmark. 117 of the mitigations were scored consistently between the five reviewers, while 33 and 46 had one or two disagreements, respectively. This produced a Fleiss' kappa of 0.53 (moderate agreement). A review of disagreements showed they most commonly arose either from a reviewer error or disagreements over subjective assessments. These failure modes are less likely when a benchmark self-scores since they have the benefit of deep knowledge about their benchmark, including non-public details.

We scored each of the 26 benchmarks using their publicly available materials. Each benchmark extended the failure mode and mitigation list. However, we did not extend the failure modes for three benchmarks (see Figure 4), which we assessed to test BenchRiskcoverage. These benchmarks involved simulators at evaluation time and similar benchmark variations requiring additional examination. For another three of the benchmarks (annotated in Table 4), the scores were entered by the benchmark authors and confirmed by BenchRisk authors. Future versions of BenchRisk will be updated to present the assertions of the benchmark authors directly affirming a mitigation has been applied and has not been invalidated through subsequent actions (e.g., sharing the data with a paying partner). We will not require benchmark authors to provide evidence of their practices – as maintainers of BenchRisk, we rely on the representations made by the benchmark authors. A research paper written by benchmark authors is not stronger evidence of a mitigation than the benchmark authors directly asserting the mitigation within the context of a risk assessment. We expect the release-time BenchRisk scores to be superseded by benchmark self-scores, as these can be more accurate than outside assessment.

# 4 Results and Discussion

The vast majority of benchmarks perform poorly in the **longevity** dimension, which we believe to result more from the goals of the benchmark authors than poor design decisions. Specifically, most of these benchmarks were produced by academic researchers who are strongly encouraged (e.g., by the NeurIPS Datasets & Benchmarks track), to publish data for reproducibility. The two outliers of AILuminate and ARC-AGI-Private are noteworthy because they do not seek to enable replication. First, the stated purpose of AILuminate includes evidencing real world decisions including

Figure 4:

| Benchmarks | Longevity | Correctness | Comprehensiveness | Consistency | Intelligibility | Mean | Min |
|---|---|---|---|---|---|---|---|
| * AILuminate | 75 | 77 | 51 | 81 | 86 | 74 | 51 |
| ARC-AGI-Private | 52 | 76 | 44 | 62 | 69 | 61 | 44 |
| WinoGrande | 4 | 68 | 33 | 86 | 65 | 51 | 4 |
| ARC-AGI-Public | 9 | 69 | 44 | 62 | 63 | 49 | 9 |
| HumanEval | 6 | 71 | 47 | 82 | 40 | 49 | 6 |
| Toxigen | 10 | 52 | 34 | 82 | 66 | 49 | 10 |
| * BBQ | 0 | 66 | 51 | 81 | 47 | 49 | 0 |
| BigBenchHard | 28 | 53 | 24 | 65 | 44 | 43 | 24 |
| GPQA | 5 | 70 | 42 | 36 | 60 | 43 | 5 |
| HellaSwag | 5 | 43 | 59 | 68 | 34 | 42 | 5 |
| BigBenchExtraHard | 28 | 54 | 24 | 65 | 39 | 42 | 24 |
| * MLC 0.5 | 21 | 34 | 48 | 36 | 66 | 41 | 21 |
| HumanitysLastExam | 11 | 41 | 52 | 37 | 61 | 40 | 11 |
| DecodingTrustToxicity | 5 | 34 | 57 | 57 | 48 | 40 | 5 |
| MMLU | 11 | 36 | 50 | 62 | 39 | 40 | 11 |
| TruthfulQA | 0 | 42 | 31 | 79 | 37 | 38 | 0 |
| Ethics | 0 | 48 | 47 | 47 | 43 | 37 | 0 |
| BigBench | 22 | 46 | 24 | 43 | 39 | 35 | 22 |
| AIRBench | 4 | 33 | 43 | 43 | 45 | 34 | 4 |
| GSM8K | 5 | 41 | 17 | 60 | 34 | 31 | 5 |
| ~~Machiavelli~~ | 0 | 22 | 31 | 65 | 37 | 31 | 0 |
| AnthropicRedTeam | 5 | 28 | 26 | 36 | 55 | 30 | 5 |
| ~~DecodingTrustPrivacy~~ | 0 | 38 | 42 | 21 | 39 | 28 | 0 |
| BOLDBias | 8 | 17 | 47 | 22 | 36 | 26 | 8 |
| RealToxicityPrompts | 0 | 12 | 59 | 20 | 35 | 25 | 0 |
| ~~Wordcraft~~ | 8 | 30 | 8 | 0 | 21 | 13 | 0 |

Figure 4: BenchRisk scores for 26 benchmarks normalized to 0–100, where no risk is mitigated at zero and all known risk is mitigated at 100. "*" indicates a BenchRisk author is among the authors of the scored benchmark. A strikethrough indicates benchmarks whose unique failure modes (e.g., simulator calibration) were deemed out of scope for the current failure mode list.

"deliver[ing] valuable insights to help enterprises deploy reliable systems that deliver business value" (ML Commons [2024]). There is no "business value" to a benchmark that is immediately saturated, which meant many of the BenchRisk longevity-related mitigations are business imperatives. For example, if the LLM-as-a-judge used for AILuminate is used by SUT developers (Failure Mode #025, "SUT developers place evaluator or other test ground truth within system chain"), then any system developer will immediately be able to score a perfect safety score by filtering all true and false safe outputs. Consequently, AILuminate adopted Mitigation 28 ("Do you refrain from making the evaluator or ground truth publicly available?") and scored highly on longevity.

The second-highest longevity benchmark, ARC-AGI-Private, was the basis for a series of competitions beginning in 2019. Competitions have distinctive practices motivated by making scores more robust to exploitation. For example, Failure Mode #015: "Prompts have known properties allowing for achieving an unrealistic (i.e., non-generalizing) performance..." has mitigation 145, "Do you avoid releasing the test set to SUT developers?" Competitions must adopt mitigation 145 to maintain competition integrity, thus they likely score highly for longevity. However, competitions often terminate on a definite timeline, so not all competition practices are consistent with benchmark longevity. All low-longevity benchmarks scored in BenchRisk would substantially increase their longevity by adopting more confidential practices.

Although we found few high-longevity benchmarks, we examined the relationship between BenchRisk longevity and how benchmark scores evolved through time. We selected all benchmarks that *1)* report a human baseline performance (i.e., we found they reported mitigation 108 requiring a human baseline) (R0bk [2025]), and *2)* are at least a year post-publication. We then found a pair of performance points for each of the remaining seven benchmarks. The first entry of the pair gives the top performance at release normalized to zero, while the second value represents the first SUT performance exceeding the human baseline. Both values are normalized so that the

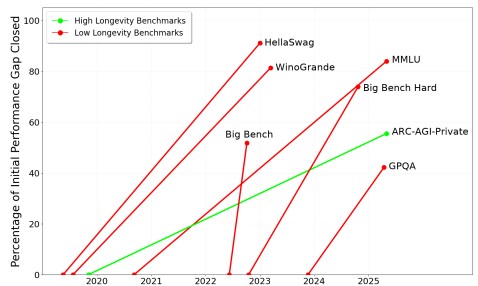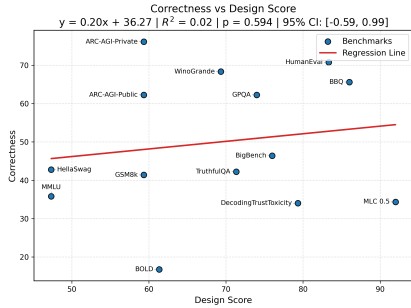

Figure 5: Exploring the properties of time-to-saturation and benchmark quality. **Left**: Time to saturation for selected benchmarks. The y-axis shows the percentage of the performance gap (between initial SOTA and a perfect score) that has been closed. 0% represents the SOTA at benchmark release; 100% represents a perfect score. Benchmarks with high BenchRisk longevity scores (green) show a slower saturation rate. **Right:** BenchRisk correctness (y-axis) and BetterBench design (x-axis) scores plotted with a regression line. The two features appear to be independent of one another.

minimum on the y-axis corresponds to the top performing SUT at the time of the benchark's release, and the maximum corresponds to a perfect score. ARC-AGI-Private presents comparatively little improvement between its release in 2019 and the middle of 2024 (See Chollet et al. [2025], Chollet [2024], Kamradt [2025] for an exploration of how the original ARC challenge has recently met its end). All the other benchmarks showed faster performance improvement, but the sample size is far too small to make conclusions. More high-longevity benchmarks are required before we can empirically build the case for BenchRisk's estimation of longevity.

We expected the longevity dimension of BenchRisk to be uncorrelated with BetterBench's measure of benchmark usability. At least one BetterBench question, "The evaluation data or generation mechanism is accessible" is in direct tension with several longevity failure modes. However, we found that BetterBench and BenchRisk may be independent of one another across all dimensions. We did not find any significant relationship between the two and provide an illustrative example in Figure 5, which shows that BetterBench's design score and BenchRisk's correctness scores are independent. The independence arises from measuring different things. BetterBench does not presume an adversarial relationship with SUT developers (correctness/longevity), less qualified users failing to read documentation (intelligibility), and safety-critical needs for a wide variety of contexts (comprehensiveness).

Poor performance on **correctness** is often associated with failure modes that may substantially bias the results in pernicious ways, such as using LLMs to produce benchmark data (Failure Mode #003: "Input prompt writers produce prompts with LLMs."). LLM-produced benchmark data *may* privilege or punish SUT scores (e.g., Panickssery et al. [2024]), but the impact is often unknown absent experimentation. Risks posed by such unknowns can be accounted for in risk management. We produced eight candidate mitigations for Failure Mode #003. While Mitigation 89 ("Are all prompts authored by the benchmark creators themselves, without using data vendors, LLMs, or crowd workers whose identities are unknown to the authors?") is rated to be the most effective and is true of some benchmarks (e.g., BBQ), most scored benchmarks use publicly available data that may be produced by an LLM (e.g., BOLD), use crowd workers that may use LLMs undisclosed to the benchmark authors (e.g., GPQA), or intentionally make use of LLMs (e.g., AILuminate, AirBench, ToxiGen). Authors may put into place Mitigation 4 ("Do you run a study on any SUT that may have been privileged during prompt generation, and compare its performance to SUTs not involved in prompt generation? If an unfair advantage is found, do you drop the LLM-generated instances?") to reduce the uncertainty.

The generality provided by LLMs challenges **Comprehensiveness** to be among the worst-performing dimensions across all benchmarks. Any benchmark could score highly on BenchRisk by limiting benchmark scope to match the coverage of the evaluation prompts. However, general-purpose models make complete coverage of their scope impossible, so benchmarks must provide statements scoping the space they rigorously cover. Failure mode #002, "The task is defined too broadly to achieve any reasonable degree of coverage over the use case," was unmitigated by 16 of the scored benchmarks.

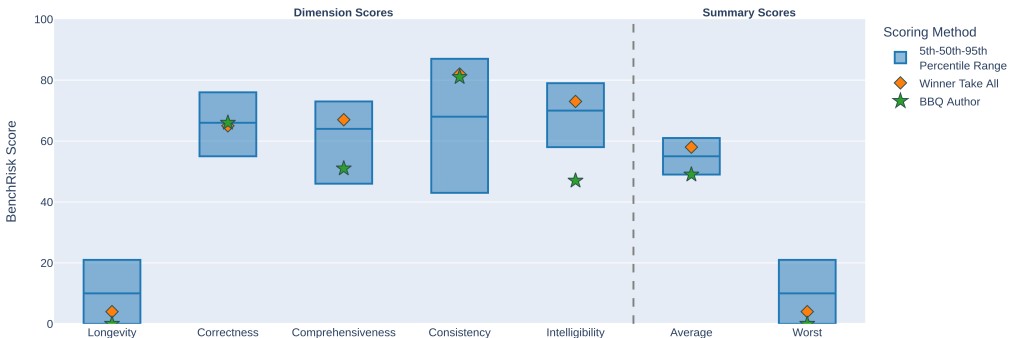

Figure 6: Agreement analysis of rater error. For each dimension, mitigations were sampled according to the probability that a single rater operating without a secondary reviewer would indicate the mitigation is present on the BBQ benchmark. We report the 5th, 50th, and 95th values of 100 Monte Carlo resamplings and a deterministic winner-take-all independent vote among the 5 raters. The winner-take-all measure shows the consensus is generally closer to the findings of the BBQ benchmark author.

Among the best performing dimensions is **consistency**, which is generally made more reliable by sampling adequate prompts within the covered space to ensure the variance of the score is adequately small. For example, Failure Mode #31 "Evaluator(s) perform poorly across all SUTs" was mitigated by all but eight of the benchmarks by mitigations 55, 98, and/or 156. 55 provides for optimizing the evaluator (e.g., LLM-as-a-judge) until it is measured to perform adequately, while mitigations 98 and 156 force benchmark requirements to make evaluation easier (e.g., requiring a bit-exact solution). If these mitigations are not in place, then the evaluator model introduces error into the benchmark estimate, and the relying user will not know whether the measured property is a probabilistic artifact.

Finally, **intelligibility** scores are moderate due to documentation practices that generally serve the research community well, but neither document nor disclaim benchmarks for real-world decision making. So while many benchmarks scored points mitigating Failure Mode #36 "Presentation without uncertainty or confidence of the scores," only AILuminate scored points for Mitigation 75, "Do you perform design studies with potential users to understand presentation requirements for benchmark outputs?" for Failure Mode #038 "User does not understand visual representation of scores."

While the other dimensions showed agreement between BenchRisk authors and a BBQ author, the intelligibility dimension presented a stronger disagreement as shown by Table 6. We believe this disparity points to a need for greater efforts to refine documentation and criteria for mitigations defined on what is communicated about the benchmarks.

## 5 Conclusion

Benchmark reliability risk evolves through time with advancing technology, science, and society. As such, the repository hosting this paper includes a collection of issue templates for publicly submitting new failure modes, mitigations, and suggested amendments to these BenchRisk components. After discussion and acceptance by the BenchRisk maintainers, amendments and additions will be announced and a new version of BenchRisk will become available for scoring.

The Anna Karenina principle (Diamond [1997]) holds success requires satisfying many conditions, while failure requires few conditions to be met. Risk management provides a means to reason about the multitude of benchmark reliability conditions and can advance the field towards greater reliability.

**Acknowledgments** The following people gave significant comments and contributions during the course of this work: Daniel Reichert, Paul Röttger, Jesse Hostetler, Rebecca Weiss, Peter Mattson, Kurt Bollacker, and Ryan Tovcimak. Thanks to Joy Braithwaite, whose discussions on applying reliability engineering methods to AI-systems informed this effort.

**Author Contributions** All authors gave considerable contributions to this work, but the character of their contributions varied. Authors marked with † assessed a large number of benchmarks and contributed to the dataset analysis. Authors marked with ‡ assessed benchmarks and contributed to the metaevaluation science of BenchRisk. Authors marked with ∘ are benchmark authors that were scored early in the BenchRisk development process and provided early advice regarding failure modes and mitigations. Authors marked with ¶ made substantial contributions to the interdisciplinary collaboration represented by this work, including a unified approach and presentation covering information risk, reliability, and statistical validity practices. All authors are permitted to reorder their names within their equivalence classes.

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

# A   Website

The community website (`https://benchrisk.ai/`) for BenchRisk includes the following.

1) **Detailed results for every scored benchmark.** This provides a wealth of contextualized details regarding BenchRisk.

2) **Failure Mode Examples.** These help explain what each failure mode is and why you should care about them.

3) **Failure mode submission processes.** This details how new failure modes are added to BenchRisk. Anyone can submit failure modes and current BenchRisk maintainers process them.

4) **Mitigation submission process.** This details how new mitigations are added to BenchRisk. Anyone can submit failure modes and current BenchRisk maintainers process them.

5) **Severity and Mitigation Coding Guide.** This details how BenchRisk's qualitative choices were arrived at by committee decision.

6) **LLM Benchmark Production Stages.** This shows how the initial set of failure modes were produced by breaking benchmark production into steps and finding potential failure modes by detailing the activities involved in each step.

7) **Glossary.** Definitions adopted for use in the production, maintenance, and application of BenchRisk.

To suggest amendments to BenchRisk as shown by Figure 7, log into GitHub and visit: `https://github.com/BenchRisk/BenchRisk/issues/new/choose`

Figure 7: A screen capture of the GitHub BenchRisk submission and amendment issue templates. This forum provides a space for BenchRisk and benchmark authors to discuss and score risks. Benchmark authors can follow the repository's announcements of accepted mitigations and failure modes and indicate whether their existing benchmarks conform to them.

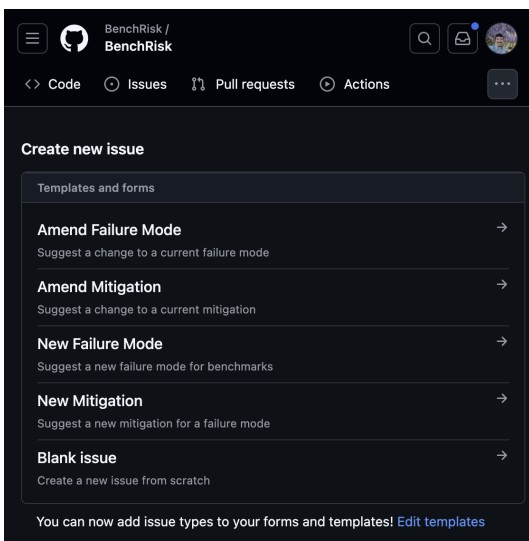

# B  BenchRisk Iterative Development

> **BenchRisk Formalized within Information Security Practices.** Established information security practices (National Institute of Standards and Technology [2012]) require explicitly specifying the following elements:
>
> 1) **Identify the purpose [of the risk assessment]:** *to identify and mitigate failure modes presenting risks to the reliability of the LLM benchmark*
>
> 2) **Identify the scope:** *to identify: a) all failure modes likely to lead a person to make a false inference about the properties of an LLM along with b) failure-mode-specific mitigations.*
>
> 3) **Identify the assumptions and constraints:** *the benchmark organization will faithfully indicate the properties of their benchmark program.*
>
> 4) **Identify the sources of information to be used as inputs:** *public statements (when scored externally) and insider knowledge (when scored by benchmark authors) about the operations of the benchmarking organization and the properties of the benchmark.*
>
> 5) **Identify the risk model and analytic approaches (i.e., assessment and analysis approaches) to be employed [during the risk assessment]:** *Divide failure modes into dimensions relevant to users interpreting a benchmark's reliability. Score those failure modes by severity and their mitigations by their capacity to reduce severity and likelihood. Total mitigated risk and explore the resulting dataset.*

BenchRisk was seeded prior to public contribution iteratively, from the ground up, by processing a series of benchmarks and research papers. At each iteration, new failure modes and mitigations were identified.

**(1) ML Commons 0.5** - LLM Product Safety Benchmark

The first benchmark scored was also the inspiration for producing BenchRisk. Released in 2024, the ML Commons 0.5 safety benchmark (Vidgen et al. [2024]) was then a state of the art benchmark representing many best practices of the era, however, when producing the benchmark for real-world safety, the benchmark authors identified a collection of risks that would lead people to a false sense of safety. We therefore collected the list of what were then merely termed "issues" along with the "fixes" adopted, where possible.

**(2) AILuminate (ML Commons 1.0)** - LLM Product Safety Benchmark

The subsequent version of the ML Commons benchmark, "AILuminate" (ML Commons [2024]), then involved 100+ researchers and engineers working together to solve the failure modes identified during the production of ML Commons 0.5. Many additional failure modes were identified over the coming months and captured into BenchRisk regardless of whether mitigations were adopted or even identified.

**(3) BBQ** - Bias Benchmark

Having twice iterated BenchRisk within the ML Commons working groups, we then turned to scoring the BBQ bias benchmark (Parrish et al. [2022]). Differences in benchmark methodology highlighted where changes to failure mode descriptions were necessary for the sake of clarity and where new mitigations would need to be introduced consistent with BBQ practices.

**(4) BetterBench** - A benchmark quality benchmark

As a matter of literature review, we coded each of BetterBench's (Reuel et al. [2024]) questions according to whether they pertain to reliability (i.e., whether a user should rely on the benchmark for decision making) and/or scientific replication (i.e., whether a benchmark is of sufficient quality for a researcher to reproduce the benchmark results with a sampling of new data). Through this analysis, we identified five new design-related failure modes not identified previously within BenchRisk along

with 19 novel BetterBench best practices as mitigations to new and previously identified failure modes.

BetterBench also includes many mitigations required to facilitate the peer review process. Peer review is a mitigation capable of addressing many varied failure modes. Therefore we associated many of the best practices identified by BetterBench that facilitate peer review as addressing a catchall failure mode of "Benchmark production failed to account for an idiosyncratic failure mode." This makes peer review serve the function of "red teaming" the benchmark, which would tend to uncover novel failure modes.

**(5,6) ARC-AGI** - A benchmark for Artificial General Intelligence

Although inspired by the reliability requirements posed by safety benchmarks, we observed that those requirements could be extended to LLM benchmarking more broadly. Consequently, we did a test run of BenchRisk on the ARC AGI (Chollet et al. [2025]) benchmark, which measures skill acquisition efficiency as a proxy for artificial general intelligence. More interesting than its intended purpose, ARC AGI has two different versions. The public version presents as a benchmark, while a "private" (i.e., tightly controlled) version presents more as a competition. The existence of the competition version means few, if any, organizations report performance on the public dataset.

The ARC authors indicate (Chollet et al. [2025]) their 2024 competitors may have reached their scores by solving prompts that were solved by at least one model 4 years prior (i.e., 49 percent of prompts were solved by at least one model in 2020). The top 3 final scores in 2024 averaged 44.5 percent, which suggest progress has not exceeded the collective capacities of systems 5 years ago. However, in December of 2024 ARC and OpenAI announced highly publicized results for a newly produced dataset drawn from the same distribution as ARC-AGI-Private, which they label as "semi-private."[1] An initial score of 88 percent was run with comparatively infinite inference-time compute budget. Since ARC challenge competitors had previously shown a relationship between solution search budget and performance, the OpenAI benchmark is not comparable to the earlier scores that operated with far less compute budget. When normalizing for compute budget, OpenAI produces an impressive but not revolutionary score of 53 percent. See Chollet [2024], Kamradt [2025] for details.

Further, the ARC authors speculate their benchmark has fallen to a new failure mode (Chollet et al. [2025]). Repeated competitions carried out over four years allowed for 10,000 benchmark evaluations. While this is not consistent with Failure Mode 46, "SUT developers can run the benchmark an unlimited number of times," 10,000 evaluations suggests, according to its authors, that ARC is now overfit. We have introduced a new candidate failure mode for addition to BenchRisk.

**(7,8,9) TruthfulQA, AIRBench, GPQA** - Other Benchmarks

Having scored 5 benchmarks and processed one research paper, the next three scored benchmarks (Lin et al. [2021], Zeng et al. [2024], Rein et al. [2025]) produced far fewer additions to the failure mode and mitigations registry. At this point, we were sufficiently confident to scale BenchRisk.

**(9-26) Scaling** - Many Additional Benchmarks

At this point we began processing many benchmarks in parallel from their publicly available documentation. We attempted to score several non-foundation model benchmarks at this point to test the boundaries of our failure mode list. While we found our definitions of failure mode concepts were robust to a broader class of LLMs, the number of additional candidate failure modes (e.g., for simulator failures) made it more expedient to leave those benchmarks to future work.

**(?) Future and Meta-metaevaluation** - Continuing improvement in response to the evolving science

When evaluating BenchRisk itself according to its own dimensions, we believe it would score highly on all dimensions except for its correctness due to the inevitable biases introduced by BenchRisk coauthors. Specifically, producing failure modes and their mitigations iteratively likely presents primacy biases because more time must be spent generating the initial failure mode and mitigation list. This introduces a variety of metaevaluation failure modes, including (1) a potential oversampling of failure modes for initial benchmarks, (2) greater coverage of initial mitigations, and (3) an under-sampling

---

[1]ARC allows "semi-private" data to be run on servers the benchmark authors do not control, but ARC undertakes efforts to ensure the evaluation set is not logged or viewed by the SUT developers. It will provide at least one additional datapoint as time passes.

of failure modes for later benchmarks. We believe (1) and (2) will tend to bias BenchRisk in favor of early benchmarks, while (3) will tend to punish early benchmarks because we are less likely to identify failure modes placed out of scope by benchmark design choices (e.g., multimodal failure modes are not possible for non-multimodal benchmarks).

Until such time that benchmark authors have an opportunity to respond to BenchRisk scores by identifying mitigations not in evidence in their public documentation, the scores should be viewed as biased lower than what could be achieved through direct participation of the benchmark authors. However, this is how risk management processes proceed: you register risks then work to address them. When an LLM benchmark is meant to evidence real world decisions, we recommend its authors adopt a risk management approach and start with the list of failure modes detailed via Appendix A's resources.

## C  Scored Benchmarks

Benchmarks were scored from the following definitive research publications augmented by viewing publicly available information on websites and blogs where necessary.

Table 2: Benchmarks used in this study and their primary references.

| Benchmark | Description — Primary citation |
|---|---|
| AILuminate | MLCommons risk & reliability benchmark v1.0 — ML Commons [2024] |
| AIRBench | Regulation-aligned AI-risk benchmark 2024 — Zeng et al. [2024] |
| AnthropicRedTeam | Red-teaming prompts for harm reduction — Ganguli et al. [2022] |
| ARC-AGI-Private | Private split of the ARC-AGI evaluation suite[2]— Chollet [2019] |
| ARC-AGI-Public | Original ARC intelligence measure (public split) — Chollet [2019] |
| BBQ | Hand-built bias benchmark for QA — Parrish et al. [2022] |
| Big Bench | General-purpose multitask evaluation suite — Srivastava et al. [2023] |
| Big Bench Hard | Hard subset of BIG-Bench tasks — Suzgun et al. [2022] |
| Big Bench Extra Hard | Extra-difficult subset of BIG-Bench — Kazemi et al. [2025] |
| BOLD Bias | Open-ended generation bias dataset — Dhamala et al. [2021] |
| DecodingTrustPrivacy | Training-set privacy slice of DecodingTrust — Wang et al. [2024] |
| DecodingTrustToxicity | Trustworthiness suite (toxicity slice) — Wang et al. [2024] |
| Ethics | Alignment with shared human-values corpus — Hendrycks et al. [2020a] |
| GPQA | Graduate-level "Google-proof" QA dataset — Rein et al. [2025] |
| GSM8K | Grade-school math word-problem set — Cobbe et al. [2021] |
| HellaSwag | Commonsense sentence-completion challenge — Zellers et al. [2019] |
| Humanity's Last Exam | Holistic AGI-oriented evaluation of LLMs — Phan et al. [2025] |
| HumanEval | Function-level code-generation accuracy corpus — Chen et al. [2021] |
| Machiavelli | Reward vs. ethical-behavior trade-off suite — Pan et al. [2023] |
| MLC 0.5 | MLCommons AI-Safety benchmark v0.5 — Vidgen et al. [2024] |
| MMLU | Massive multitask language-understanding exam — Hendrycks et al. [2020b] |
| RealToxicityPrompts | Prompt set for neural toxic-degeneration tests — Gehman et al. [2020] |
| ToxiGen | Adversarial & implicit hate-speech detection dataset — Hartvigsen et al. [2022] |
| TruthfulQA | Benchmark for truthful question-answering — Lin et al. [2021] |
| WinoGrande | Large-scale adversarial Winograd-style coreference test — Sakaguchi et al. [2019] |
| Wordcraft | Interactive story-writing environment — Jiang et al. [2020] |

