# OpenReview forum: "Risk Management for Mitigating Benchmark Failure Modes: BenchRisk"
_NeurIPS.cc/2025/Datasets_and_Benchmarks_Track — NeurIPS 2025 Datasets and Benchmarks Track poster_

### Official Review · Reviewer_1CkG · 2025-06-01

**Rating:** 4
**Confidence:** 3

**Summary:**

This paper proposes to evaluate the risk of using specific benchmarks when evaluating the models. They provide a platform and evaluated a range of commonly used benchmarks for their longevity, correctness, comprehensiveness, consistency and intelligibility.

**Dataset Code Accessibility:**

Partly

**Ethical Considerations:**

No, there are no or only very minor ethics concerns

**Final Justification:**

The authors addressed most of my concerns regarding writing, but there are remaining limitations that prevents me from giving a higher score.

**Limitations Weaknesses:**

1. I found the paper difficult to read. I am looking for the following details but did not find:
  - The risks and mitigations seem to be all subjective. How do the authors make sure the consistency in scoring?
  - There is no mathematical fundation supporting the computation of Algorithm 1. Why should we compute the score in that way?
  - The authors mentioned "The complete set of risks and mitigations are available via Appendices A and B", but I did not find them. It is actually contradictory since the mitigation for different benchmark should be different, but if you use different mitigation (i.e. different $M_{d,f}$) for different benchmark, then the scores are not directly comparable and completely depend on whether the mitigations in $M_{d,f}$ are effective or not.

2. No error bar or standard deviations reported for the scores. How reliable are they? What are the possible variations in those scores?

**Strengths Contributions:**

This work systematically looks at the possible risks with multiple aspects of possible failure modes. This will serve as a guideline of making future benchmarks or reforming existing benchmarks.

---

> ### Author Rebuttal · Authors · 2025-07-30
>
> We thank you for your thoughtful feedback and constructive suggestions. We appreciate the time and effort invested in reviewing our work. Below, we address your concerns raised in turn. If you believe we have adequately addressed your concerns, we ask that you revise your rating, as there is a wide disparity in scores among our reviewers, or let us know if we should address these points any further.
>
> > "This will serve as a guideline of making future benchmarks or reforming existing benchmarks."
>
> While expressed as a strength of the paper, we would like to clarify that we do not believe some mitigations we identify should be adopted for benchmarks produced for scientific purposes (i.e., as opposed to evidencing real world decisions from benchmark results). For example, a benchmark that is downloadable by researchers may advance the state of the art in LLM capabilities more quickly than a benchmark that is more private for the purpose of protecting its longevity. Therefore, we do not believe that all benchmarks should be reformed per our mitigations, but those that seek to inform real world decisions should seek to mitigate the failure modes we identify.
>
> > "1. I found the paper difficult to read. I am looking for the following details but did not find:
>
> > "The risks and mitigations seem to be all subjective. How do the authors make sure the consistency in scoring?"
>
> Line 188 through 192 read: "All scores presented within this work were subject to a primary and secondary reviewer, who discussed and eventually reached agreement on the appropriateness of affirming a mitigation given publicly known information about each benchmark. Reaching agreement sometimes involved clarifying descriptions of failure modes and their mitigations, which were captured and applied for all scores."
>
> The points of subjectivity include the list of failure modes, mitigations, and whether the mitigations have been faithfully implemented. Our approach in each instance was to drive towards consensus and provide an open process for submission and amendment. The identification of failure modes and their mitigations is a standard systems engineering approach for improving reliability. In reliability engineering, the identification of new failure modes and mitigations never ends. The process of continual identification is what can feed ever-increasing improvements in reliability. Through this research paper and extensive application of BenchRisk, we populated an initial list representative of a wide range of benchmarks. With this starting point, we can now invite the community to further refine to a shared consensus as indicated via the website linked in Appendix A.
>
> Methodologically, this is the hardest element of this work to "get right" and we encourage your further review of supporting materials and other reviews/rebuttals for further discussion indicating how we pursued this work.
>
> > There is no mathematical fundation supporting the computation of Algorithm 1. Why should we compute the score in that way?
>
> The base calculation of risk score (likelihood * severity) has many decades of precedent. In building on this precedent, we cite many risk management and reliability processes that implement our algorithm without mathematical formalization. In this work, we are not seeking to reinvent these widely adopted processes. BenchRisk's first innovation is applying those processes within a benchmarking context to provide for a metaevaluation. Thus we inherit a rich history of empirical evidence for our approach. BenchRisk's other innovation is the identification and description of failure modes that contribute to risk.
>
> > The authors mentioned "The complete set of risks and mitigations are available via Appendices A and B", but I did not find them. It is actually contradictory since the mitigation for different benchmark should be different, but if you use different mitigation (i.e. different  Md,f) for different benchmark, then the scores are not directly comparable and completely depend on whether the mitigations in Md,f are effective or not.
>
> We believe this was a miscommunication on our part. Appendix A is a list of contents linked to at benchrisk.ai containing these materials. We will edit the text to make it clear that the appendices do not themselves contain the full list of risks and mitigations. As the website allows us to use interactive tables, we feel this is a more intuitive presentation of the long list of mitigations considered, which do indeed often differ for different benchmarks.
>
> In scoring, we reached an expert consensus of mitigation effectiveness among coauthors, though you raise a valuable point that two different mitigations (or even the same mitigation applied to very different benchmarks) may not be directly comparable in the degree to which the risk is mitigated. We minimize this tension by introducing additional mitigations for benchmarks when we believe a benchmark may be unfairly punished or advantaged in the BenchRisk scores. For instance, Failure Mode 9 is labeled as "Adversarial prompt bulking (increasing the number of prompts by multiplying them by the number of tactics," but most benchmarks are asking questions unrelated to adversarial robustness and those benchmarks may underperform relative to benchmarks more at risk to Failure Mode 9. Thus we added Mitigation 165, which asks: "Is the task defined in a way that does not examine adversarial performance, and are adversarial prompts excluded?" The first clause clarifies that the mitigation is only applicable for certain benchmark types, while the second clause affirms the property that is implied by the first clause.
>
> Finally, the list of mitigations and their impacts on failure modes will never be exhaustive. Reliability engineering is an iterative process—one that involves continuously identifying, modeling, and refining failure modes and their mitigations. To accommodate this, we designed BenchRisk with a flexible framework for introducing and updating failure modes and mitigations. While BenchRisk serves as a valuable tool for analyzing current benchmarks, we anticipate it will evolve alongside advances in metrology science and our growing understanding of benchmark reliability.
>
> > "2. No error bar or standard deviations reported for the scores. How reliable are they? What are the possible variations in those scores?"
>
> The amount of effort required to score each benchmark from publicly available materials is immense. Unfortunately, we lack the resources necessary to produce error bars for Figure 3 as that would require independently scoring each benchmark multiple times. We performed and reported a limited study of interrater agreement and found moderate agreement on a single benchmark. As discussed in the paper, we believe this is a strong score for qualitative coding work that requires a degree of guessing about what the benchmark authors are expressing in their research papers. When benchmark authors self-score, there is no doubt about their benchmarking practices.
>
> While we lack sufficient resources to produce rater agreement error bars on a per-benchmark basis, we can and will report the uncertainty of each dimension score (i.e., intelligibility, consistency, correctness, longevity, comprehensiveness) arising from rater disagreements on the BBQ benchmark and will update the visual representation to include the uncertainty registered for BBQ. We debated whether such numbers would mislead BenchRisk users since they all arise from the BBQ benchmark, but the risk of BenchRisk users being misled by the absence of any form of error bars exceeds the risk introduced by the BBQ error estimate not applying to other benchmarks. We will appropriately characterize the limitations of this error estimate within the camera ready version of the paper.
>
> ...
>
> Thank you again for your time and consideration. We are happy to engage further on any of these points during the discussion period.

---

> > ### Comment · Reviewer_1CkG · 2025-08-02
> >
> > Thank you for the clarification. It addresses most of my concerns (which are mainly in writing). I will adjust my rating to 4 but the remaining limitations including error bars and the direct comparison issues stops me from giving a higher score.

---

> > > ### Author Response · Authors · 2025-08-04
> > >
> > > Thank you for your thought and input and for updating the score. We will redouble our efforts on the outstanding points you highlight.

---

### Official Review · Reviewer_YXpA · 2025-06-29

**Rating:** 5
**Confidence:** 3

**Summary:**

This paper builds a benchmark to measure the reliabilities of benchmarks. Specifically, the authors adopted the National Institute of Standards and Technology 's risk management process to analyse 26 popular benchmarks. Then they identified the potential failure modes and corresponding mitigation strategies. As the demonstration, the authors provided the BenchRisk scores for these 26 benchmarks, and how to understand each dimension.

**Dataset Code Accessibility:**

Yes

**Ethical Considerations:**

No, there are no or only very minor ethics concerns

**Final Justification:**

The authors promise to add more examples to increase readability. And I am satisfied about the authors' response on how to handle the uncertainty or rater agreement errors, and the trust on self-reported scores from the benchmark author side.

Overall, I think this paper targets on an important research problem and builds a pipeline to assist decision-making when choosing a benchmark. Meanwhile, they promise to have evolving efforts (webpages, GitHub repos, etc.) to benefit the community.  I recommend to accept the paper.

**Limitations Weaknesses:**

From my perspective, there is no clear technical weakness in this paper. But I do find the paper difficult to read in the first impression. This paper is not a typical AI paper or technical report. In the writing, the authors use large space to utilize the terminologies and languages in NIST to conceptualize the pipeline. However, several concrete examples are absent in the main body.

I think it is better to include some concrete examples of failure modes, adopted mitigation, how to determine the values of $m_s$ and $m_l$ when the authors first introduce these terms.

The authors mentioned that different assessors may have disagreements when scoring the benchmark. I think it is better to include the error bars in Figure 3 if possible. Moreover, if the authors report self-scores, should the audience fully trust the scores. Is there any procedure to increase the trust of benchmark scores?

**Strengths Contributions:**

1. The paper is clearly very novel with new aspect to benchmark benchmarks and new design pipeline.

2. The problem researched in this paper is very important. With the popularity of LLMs, various benchmarks were proposed. However, whether these benchmarks are reliable is not systematically evaluated. This paper fills in the gap and provides a score to measure the risks.

3. Together with the paper, the authors provide infrastructure, GitHub repo, and an evolving website to measure the benchmarks beyond the ones mentioned in this paper. I think this paper could be impactful to the AI community.

---

> ### Author Rebuttal · Authors · 2025-07-30
>
> We thank you for your thoughtful feedback and constructive suggestions. We appreciate the time and effort invested in reviewing our work, and we believe the comments show attention to the purpose and details of our work. Below, we address the two concerns raised in turn.
> > "...several concrete examples are absent in the main body."
>
> > "I think it is better to include some concrete examples of failure modes, adopted mitigation, how to determine the values of $m_s$ and $m_l$ when the authors first introduce these terms."
>
> We agree with this criticism and intend to use the entirety of the extra page afforded to accepted papers (if indeed we are accepted) to provide these examples.
> > "The authors mentioned that different assessors may have disagreements when scoring the benchmark. I think it is better to include the error bars in Figure 3 if possible."
>
> The amount of effort required to score each benchmark from publicly available materials is immense. Unfortunately, we lack the resources necessary to produce error bars for Figure 3 as that would require independently scoring each benchmark multiple times. We performed and reported a limited study of interrater agreement and found moderate agreement on a single benchmark. As discussed in the paper, we believe this is a strong score for qualitative coding work that requires a degree of guessing about what the benchmark authors are expressing in their research papers. When benchmark authors self-score, there is no doubt about their benchmarking practices.
>
> While we lack sufficient resources to produce rater agreement error bars on a per-benchmark basis, we can and will report the uncertainty of each dimension score (i.e., intelligibility, consistency, correctness, longevity, comprehensiveness) arising from rater disagreements on the BBQ benchmark and will update the visual representation to include the uncertainty registered for BBQ. We debated whether such numbers would mislead BenchRisk users since they all arise from the BBQ benchmark, but the risk of BenchRisk users being misled by the absence of any form of error bars exceeds the risk introduced by the BBQ error estimate not applying to other benchmarks. We will appropriately characterize the limitations of this error estimate within the camera ready version of the paper.
>
> > "Moreover, if the authors report self-scores, should the audience fully trust the scores. Is there any procedure to increase the trust of benchmark scores?"
>
> This is a real concern for us moving forward, however, we have mitigations. ;)
> Among the mitigations are:
> 1. **Intrinsic duty:** A sense of professional duty shared by colleagues advancing the state of knowledge about AI systems. All scientific progress relies on a good deal of trust that colleagues are behaving in a trustworthy way.
> 2. **Extrinsic motivation:** When people rely on your representations relevant to safety, you bear liability for any fraudulent misrepresentation. For instance, if an airline relies on a benchmark when deploying a chatbot, if/when the airline loses money because the benchmark did not implement a mitigation they claimed is in place, then they may be liable for commercial misrepresentation for all the losses incurred by the airline.
>
> These mitigations may clarify the purpose to which researchers are publishing benchmarks -- why expose yourself to the criticism of your peers and financial liabilities by lying on a metaevaluation if you are not attempting to evidence real world decisions? A benchmark produced for research purposes does not require high longevity or coverage of the complete space to be groundbreaking.

---

> > ### Comment · Reviewer_YXpA · 2025-08-02
> >
> > Thank you for your response. I will keep my original rating.

---

### Official Review · Reviewer_FJad · 2025-06-30

**Rating:** 4
**Confidence:** 2

**Summary:**

BenchRisk inspects 26 popular benchmarks, enumerated 57 failure modes and 196 mitigations, then produced a quantitative “reliability score” along five dimensions—comprehensiveness, intelligibility, consistency, correctness and longevity.

**Additional Feedback:**

See limitations

**Dataset Code Accessibility:**

Yes

**Dataset Code Comments:**

Availability: All failure-mode definitions, mitigation lists, and benchmark-level scores live in the public GitHub (BenchRisk/BenchRisk) under an open licence.

**Ethical Comments:**

No ethical comments

**Ethical Considerations:**

No, there are no or only very minor ethics concerns

**Final Justification:**

Many thanks to the author rebuttals. I agree with the other reviewers that this paper should be accepted. I have updated my score to indicate acceptance.

**Limitations Weaknesses:**

1. The main weakness of this work is unclear framing. What do we get out of BenchRisk? Does the BenchRisk assessment provide conclusive evidence that we should not generate benchmarks when they might rank low on BenchRisk? The delivery of the paper makes these questions very unclear.
2. Risk-score formula lacks theoretical or empirical grounding. How do we know that the risk score formula is correct?
3. Rubric structurally penalises openness, which may clash with NeurIPS dataset policy. Does BenchRisk encourage closed benchmarks like ARC-Private? If so doesn't that violate the NeurIPS dataset policy?
4. Finally, this paper clearly suffers from bad writing. For example, the caption of Figure 2 should be below the figure. Generally papers that do not follow the NeurIPS style may risk getting desk rejected.

**Strengths Contributions:**

1. Creates meta-benchmark scores for 26 datasets, providing an additional axis of comparison across today’s most-used LLM benchmarks
2. Open-source tooling & data (GitHub repo + interactive website) enables community re-scoring, new failure-mode proposals, and transparency
3. Algorithm-1 scoring rubric details a clear and reproducible computation of per-dimension risk reduction

---

> ### Author Rebuttal · Authors · 2025-07-30
>
> We thank you for your thoughtful feedback and constructive suggestions. We appreciate the time and effort invested in reviewing our work, and we believe the comments show attention to the purpose and details of our work. Below, we address the four concerns raised in turn.
> 1. "The main weakness of this work is unclear framing. What do we get out of BenchRisk? Does the BenchRisk assessment provide conclusive evidence that we should not generate benchmarks when they might rank low on BenchRisk? The delivery of the paper makes these questions very unclear."
>
> The target audience of this paper is researchers aiming to produce benchmarks that evidence real world decisions. While we cite some research papers with this goal in the paper (e.g., AIRBench [1] and AILuminate [2]), most research papers are not aiming to have people rely on their work when deciding whether a particular product is safe (or not) for their use. However, the extent that specific benchmarks are not prepared to evidence real world decisions is often not known to prospective benchmark users. This is a consequence of the common difficulty of resolving multiple purposes to which a benchmark may be authored, and not a shortcoming specific to the research or the real world use case. Thus we aim to provide benchmark authors with a collection of candidate mitigations and their users with information about how well the implemented set of mitigations reduce the risk of inappropriately relying on a benchmark (e.g., when a benchmark moves in-distribution for future LLMs). The concern you raise that people may interpret a low score on BenchRisk to mean the benchmark should not be used *at all* is a serious one, and we will update the paper and website with more clear mentions of BenchRisk’s focus on *real-world decision making* as well as add explicit text indicating that low-scoring benchmarks may still be appropriate for other use cases.
>
> 2. "Risk-score formula lacks theoretical or empirical grounding. How do we know that the risk score formula is correct?"
>
> The base calculation of risk score (likelihood * severity) has many decades of precedent. In building on this precedent, we cite many risk management and reliability processes that implement our algorithm without mathematical formalization. In this work, we are not seeking to reinvent these widely adopted processes. In addition to identifying and describing failure modes with mitigations, our innovation is applying those processes within a benchmarking context to provide for a metaevaluation. Thus we inherit a rich history of empirical evidence for our approach. The first few paragraphs in Section 3 provide the grounding for risk scoring, so we will edit the text in that section to make it clear that risk scoring, as we’re applying it, comes directly from the established work cited there.
>
> 3. "Rubric structurally penalises openness, which may clash with NeurIPS dataset policy. Does BenchRisk encourage closed benchmarks like ARC-Private? If so doesn't that violate the NeurIPS dataset policy?"
>
> Most papers submitted to the NeurIPS benchmark track are produced for scientific purposes rather than real world decision making purposes, but people are still making real world decisions on the basis of these research results. We would like to reiterate (and we’ll be very clear about this in updates to the paper) that a low score on BenchRisk does not indicate a fundamental flaw in the research or creation around a benchmark, it means that there are risks associated with using that benchmark for deployment decisions in real world contexts. We believe both open and closed benchmarking have their place in the community. Further, leading NeurIPS benchmarks exhibiting low longevity continue to be strong evidence of unsolved problems in machine learning even when a saturated benchmark does not indicate a problem has been solved in its full real world complexity (i.e., generality).
>
>
> 4. "Finally, this paper clearly suffers from bad writing. For example, the caption of Figure 2 should be below the figure. Generally papers that do not follow the NeurIPS style may risk getting desk rejected."
>
> Thank you for pointing out the placement issue with Figure 2. We moved the caption to its proper location and will re-read the style guide to ensure 100 percent compliance prior to any potential camera ready version.
>
> ...
>
> Thank you again for your time and attention. We are happy to engage further on any of these points during the discussion period. If you believe we have adequately addressed your concerns, we ask that you revise your rating, as there is a wide disparity in scores among our reviewers, or let us know if we should address these points any further.
>
> [1] Yi Zeng, Yu Yang, Andy Zhou, Jeffrey Ziwei Tan, Yuheng Tu, Yifan Mai, Kevin Klyman, Minzhou Pan, Ruoxi Jia, Dawn Song, Percy Liang, and Bo Li. Air-bench 2024: A safety benchmark based on risk categories from regulations and policies. 2024.
>
> [2] ML Commons. AILuminate benchmark. https://<redacted to comply with rebuttal requirements>, 2024. [Accessed 28-01-2025].

---

### Official Review · Reviewer_wVCc · 2025-07-03

**Rating:** 5
**Confidence:** 3

**Summary:**

This paper introduces BenchRisk, a metaevaluation framework for assessing the reliability of LLM benchmarks. The authors apply risk management principles from reliability engineering to identify 57 potential failure modes across 26 popular LLM benchmarks and propose 196 mitigation strategies. The framework evaluates benchmarks across five dimensions: comprehensiveness, intelligibility, consistency, correctness, and longevity. Higher BenchRisk scores indicate that benchmark users are less likely to reach incorrect or unsupported conclusions about an LLM's capabilities or safety.

**Additional Feedback:**

Perhaps consider designing empirical studies comparing deployment decisions made with high vs. low BenchRisk-scored benchmarks to validate the framework's real-world utility.

**Dataset Code Accessibility:**

Yes

**Ethical Considerations:**

No, there are no or only very minor ethics concerns

**Limitations Weaknesses:**

1. While the paper mentions inter-rater reliability (Fleiss' kappa of 0.53), this represents only moderate agreement. The scoring system could heavily rely on expert judgment without much empirical validation. The authors acknowledge this limitation but could strengthen their approach with more rigorous validation methods.
2. The longevity analysis provides only limited evidence from ARC-AGI comparisons. The small sample size might not be able to draw meaningful conclusions about the relationship between BenchRisk longevity scores and actual benchmark degradation over time.

**Strengths Contributions:**

1. This paper introduces a structured risk management framework, BenchRisk, for LLM benchmarking and provides a practical approach to mitigate benchmark failures. The proposed approach has detailed scoring mechanisms, allowing clear comparison and assessment of benchmark reliability.
2. The paper effectively motivates the need for benchmark reliability assessment, particularly given the increasing use of benchmarks for real-world deployment decisions and safety assessments.
3. The paper did extensive analysis covering 26 popular benchmarks, clearly illustrating prevalent failure modes.
4. This work offers an open-source tool and repository allowing for continuous improvement and collaboration within the community.

---

> ### Author Rebuttal · Authors · 2025-07-30
>
> We thank you for your thoughtful feedback and constructive suggestions. We appreciate the time and effort invested in reviewing our work, and we believe the comments show attention to the purpose and details of our work. Below, we address the two concerns raised in turn.
>
> > "While the paper mentions inter-rater reliability (Fleiss' kappa of 0.53), this represents only moderate agreement. The scoring system could heavily rely on expert judgment without much empirical validation. The authors acknowledge this limitation but could strengthen their approach with more rigorous validation methods."
>
> We agree with this characterization of our work -- it does rely heavily on expert judgement as indicated on lines 68, 69, and 167. The subtext of your comment may be, "is the expert judgement too subjective to be relied upon?" We would like to separate this weakness into two parts.
>
> 1) The paper of today. Most of the benchmarks we scored were not scored by the authors of the benchmarks, which means we needed to follow a rule affirming mitigations when we believe the benchmark author would likely seek to make such a claim. We know this is noisy and subjective in nature, thus none of the benchmarks we scored rely on a single rater. All scores were reviewed by a second rater and disagreements were discussed and resolved jointly. Therefore, the subsequent inter-rater agreement experiment, which did not benefit from secondary review or a discussion period, represents a floor for inter-rater reliability rather than BenchRisk's expected inter-rater reliability. When examining the failure modes explaining disagreements, we found they typically relate to knowledge that is discoverable from publicly available sources other than the definitive research paper. Our camera-ready version of the paper will include an additional table of where BenchRisk itself may fail to reliably express mitigations that are in effect.
> 2) The community of tomorrow. Our ambition is to support benchmark authors self-scoring their own benchmarks. This eliminates all uncertainty regarding the methods and practices adopted by the benchmark, but may introduce biases where the benchmark authors take more liberal interpretations of mitigation definitions than is warranted. This is a question we plan to interrogate in the presence of more data from benchmark authors. We believe publishing a peer reviewed research paper with the methodology is an important step to subsequently publishing a self-scored study.
>
> > "The longevity analysis provides only limited evidence from ARC-AGI comparisons. The small sample size might not be able to draw meaningful conclusions about the relationship between BenchRisk longevity scores and actual benchmark degradation over time."
>
> We agree with this characterization. We hope that the BenchRisk paper might serve to register a hypothesis that becomes testable as the history of LLM benchmarking continues to unfold. We expect to revisit this question in two years.
>
> > "Perhaps consider designing empirical studies comparing deployment decisions made with high vs. low BenchRisk-scored benchmarks to validate the framework's real-world utility."
>
> Agreed! Empirical studies would be valuable future work, and we’ll incorporate concrete examples of how BenchRisk can be used in this way in the paper.

---

> ### Comment · Reviewer_wVCc · 2025-08-04
>
> Thanks for the further explanation. I will keep my score.

---

### Note · Authors · 2025-08-13

We introduce BenchRisk, a novel meta-evaluation framework that applies established risk management principles to assess the reliability of LLM benchmarks. The work was received positively by reviewers (wVCc, YXPA, and 1CkG), who recognize that it addresses a "very important" (YXPA) and timely problem, given the increasing reliance on benchmarks to inform real-world deployment and safety decisions (wVCc).

Reviewers highlight the paper conducts an **extensive analysis,** covering 26 popular benchmarks and identifying 57 failure modes with 196 mitigations (wVCc, FJad). The approach is considered **"clearly very novel"** (YXPA) and provides a **"structured risk management framework"** with a **"detailed scoring mechanism"** (wVCc). All reviewers recognize the significant **accompanying infrastructure,** including open-source tooling, datasets, and an interactive website (wVCc, FJad, YXPA). This infrastructure facilitates **transparency** and enables **continuous community collaboration** and improvement.

We constructively addressed the limitations during the rebuttal:

* **Subjectivity and Inter-Rater Reliability (IRR):** R-wVCc and R-1CkG noted the reliance on expert judgment and moderate IRR. We clarified that this score represents a floor, as all scores involved a rigorous consensus process.
* **Readability and Examples:** R-YXPA and R-1CkG requested clarification and examples. We committed to improving clarity and utilizing an extra page for examples.
* **Error Bars:** R-YXPA and R-1CkG requested error bars for the scores. While acknowledging resource requirements preventing comprehensive production of error bars, we committed to reporting uncertainty estimates derived from the IRR study.
* **Framing and Theoretical Grounding:** R-FJad and R-1CkG questioned the framing and mathematical foundation of the risk score formula. We successfully addressed these points by clarifying the scope—BenchRisk assesses fitness for _real-world deployment decisions,_ not necessarily general scientific utility—and by grounding the risk calculation in decades of established precedent in reliability engineering.

During the discussion phase R-wVCc, R-YXPA, and R-1CkG engaging positively, leading R-1CkG to raise their score. While R-FJad did not participate in the discussion phase, we provided thorough responses. Overall, BenchRisk provides a timely, systematic, and impactful framework, offering valuable open-source resources to improve the reliability of LLM benchmarking.

---

### Decision · Program_Chairs · 2025-09-18

**Decision:**

Accept (poster)

**Comment:**

Summary:
The paper evaluates 26 major benchmarks through this evaluative “lens” of five different types of “failure” modes: comprehensiveness, intelligibility, consistency, correctness, and longevity.  They find 12 present significant risk within one or more of the five scored dimensions :comprehensiveness, intelligibility, consistency, correctness, and longevity.  The authors also provide extensive mitigation strategies.

In contrast the to the 2024 NeurIPS paper “BetterBench” which outlines standards for what a benchmark should include, this paper instead focuses on the potential risks of relying on a benchmark for its stated purpose.

Reviewers provided scores (5,5,4,4) where one of the “4” raters did not significantly engage and the other “4” is raised from a lower score and they say that not all of their concerns were addressed and that this prevents them from raising their score to a 5.

Strengths:
- The paper effectively motivates the need for benchmark reliability assessment, particularly given the increasing use of benchmarks for real-world deployment decisions and safety assessments.
- The proposed approach has detailed scoring mechanisms, allowing clear comparison and assessment of benchmark reliability.
- The paper did extensive analysis covering 26 popular benchmarks, clearly illustrating prevalent failure modes.
- This work offers an open-source tool and repository allowing for continuous improvement and collaboration within the community.

Weaknesses:
- Three of the reviewers found the paper difficult to read
- Two of the reviewers were concerned that the scores had no error bars
- Two of the reviewers expressed concern that the scoring system was qualitative and one raised concern that the assessment that the authors propose requires expert judgement and may be difficult for others to implement.  They raise the concern that even with expert raters, inter-rater reliability (Fleiss' kappa of 0.53), this represents only moderate agreement (this weakness is acknowledged)
- One reviewer raised that the longevity analysis provides only limited evidence from ARC-AGI comparisons. The small sample size might not be able to draw meaningful conclusions about the relationship between BenchRisk longevity scores and actual benchmark degradation over time.
- The paper does have some less standard formatting such as captions on top of and to the side of figures.  One reviewer was suggesting desk reject because of this but the authors are promising to correct these formatting irregularities in the final version if accepted.